# Soil water regulates the control of photosynthesis on diel hysteresis between soil respiration and temperature in a desert shrubland

Ben Wang[1,2], TianShan Zha[*1], Xin Jia[1,2], JinNan Gong[2], Charles Bourque[3], Wei Feng[1], Yun Tian[1], Bin Wu[1], YuQing Zhang[1], Heli Peltola[2]

[1]Yanchi Research Station, School of Soil and Water Conservation, Beijing Forestry University. Beijing 100083, PR China.

[2]School of Forest Sciences, University of Eastern Finland. PO Box 111, FIN-80101 Joensuu, Finland.

[3]Faculty of Forestry and Environmental Management, University of New Brunswick, PO Box 4400, 28 Dineen Drive, Fredericton, New Brunswick, E3B 5A3, Canada

*Correspondence to*: TianShan Zha (tianshanzha@bjfu.edu.cn)

**Abstract.** Explanations for the occurrence of hysteresis (asynchronicity) between diel soil respiration ($R_s$) and soil temperature ($T_s$) have evoked both biological and physical mechanisms. The specifics of these explanations, however, tend to vary with the particular ecosystem or biome being investigated. So far, the relative degree of control of biological and physical processes on hysteresis is not clear for drylands. This study examined the seasonal variation in diel hysteresis and its biological control in a desert-shrub ecosystem in northwest (NW) China. The study was based on continuous measurements of $R_s$, air temperature ($T_a$), temperature at the soil surface and below ($T_{surf}$ and $T_s$), volumetric soil water content ($SWC$), and photosynthesis in a dominant desert shrub (i.e., *Artemisia ordosica*) over an entire year in 2013. Trends in diel $R_s$ were observed to vary with $SWC$ over the growing season (April to October). Diel variations in $R_s$ were more closely associated with variations in $T_{surf}$ than with photosynthesis as $SWC$ increased, leading to $R_s$ being in phase with $T_{surf}$, particularly when $SWC > 0.08$ m$^3$ m$^{-3}$ (ratio of $SWC$ to soil porosity = 0.26). However, as $SWC$ decreased below 0.08 m$^3$ m$^{-3}$, diel variations in $R_s$ were more closely related to variations in photosynthesis, leading to pronounced hysteresis between $R_s$ and $T_{surf}$. Incorporating photosynthesis into a $Q_{10}$-function eliminated 84.2% of the observed hysteresis, increasing the overall descriptive capability of the function. Our findings highlight a high degree of control by photosynthesis and $SWC$ in regulating seasonal variation in diel hysteresis between $R_s$ and temperature.

## 1 Introduction

Diel hysteresis (asynchronicity) between soil respiration ($R_s$) and soil temperature ($T_s$) is widely documented for forests (Tang et al., 2005; Gaumont-Guay et al., 2006; Riveros-Iregui et al., 2007; Stoy et al., 2007; Vargas and Allen, 2008; Jia et al., 2013), grasslands (Carbone et al., 2008; Barron-Gafford et al., 2011), and desert ecosystems (Wang et al., 2014; Feng et

al., 2014). Diel hysteresis, which appears as an elliptical loop in the relationship between $R_s$ and $T_s$, is difficult to model with
theoretical functions, such as the $Q_{10}$, Lloyd-Taylor, Arrhenius, or van't Hoff functions (Lloyd and Taylor, 1994; Winkler et
al., 1996; Davidson et al., 2006; Phillips et al., 2011; Oikawa et al., 2014), leading to an inadequate understanding of
temperature-sensitivity in $R_s$ (Gaumont-Guay et al., 2008; Phillips et al., 2011; Darenova et al., 2014). Therefore, in order to
accurately predict soil carbon dioxide ($CO_2$) fluxes and their responses to climate change, it is necessary to understand the
biophysical mechanisms that have a role in controlling seasonal variation in diel hysteresis.
Over decades of research, two main processes have been reported to relate to diel hysteresis between $R_s$ and $T_s$. One is
associated with the physical processes of heat and gas transport in soils (Vargas and Allen, 2008; Phillips et al., 2011; Zhang
et al., 2015). Generally, soil $CO_2$ fluxes are measured at the soil surface, and are related to temperatures in the soil. Transport
of $CO_2$-gas to the soil surface takes time to occur, which may cause delays to appear in observed respiration rates, causing
hysteretic loops to form between $R_s$ and $T_s$ (Zhang et al., 2015). The other is associated with the biological process of
photosynthate supply (Tang et al., 2005; Kuzyakov and Gavrichkova, 2010; Vargas et al., 2011; Wang et al., 2014). Beyond
the control of temperature, soil $CO_2$ fluxes have been associated with plant photosynthesis. Photosynthesis usually peaks at
midday (e.g., 11:00-13:00), providing substrate for belowground roots and rhizosphere-microbe respiration, but oscillates out
of phase with $T_s$, usually peaking in the afternoon (e.g., 14:00-16:00). Such influences of current photosynthesis could lead
to the formation of hysteretic loops in the relationship between $R_s$ and $T_s$. These studies highlight the need to consider the
inherent role of photosynthesis for a more accurate interpretation of $R_s$ (Tang et al., 2005; Kuzyakov and Gavrichkova, 2010;
Vargas et al., 2011). Physical and biological processes that relate to substrates and production-transport of carbon (C) in
plants and soils are not mutually exclusive and both likely play crucial roles in affecting diel variation in $R_s$ (Stoy et al., 2007;
Phillips et al., 2011; Zhang et al., 2015; Song et al., 2015a, b).
Diel hysteresis between $R_s$ and $T_s$ has been shown to vary seasonally with soil water content ($SWC$; Tang et al., 2005;
Riveros-Iregui et al., 2007; Carbone et al., 2008; Vargas and Allen, 2008; Ruehr et al., 2009; Wang et al., 2014). However,
the influences of $SWC$ on diel hysteresis are not uniform. Based on the Millington-Quirk model, high $SWC$ blocks $CO_2$-gas
and thermal diffusion (Millington and Quirk, 1961), resulting in large hysteresis loops (Riveros-Iregui et al., 2007; Zhang et
al., 2015). In contrast, other studies have reported that low $SWC$ and high water vapor pressure deficits ($VPD$) can promote
partial stomata closure, which leads to higher photosynthesis in the morning (e.g., 9:00-10:00) and supressed photosynthesis
in mid-afternoon, leading to pronounced hysteresis during dry periods (Tang et al., 2005; Vargas and Allen, 2008; Carbone
et al., 2008; Wang et al., 2014). Clearly to understand the causes of diel hysteresis, the role of $SWC$ needs to be closely
evaluated.
Drylands cover a quarter of the earth's land surface and play an important role in the global C cycle (Safriel and Adeel,
2005; Austin, 2011; Poulter et al., 2014). Many studies in forest ecosystems are based on the application of physical soil $CO_2$
and heat transport models and evaluate the influences of $SWC$ on $CO_2$-gas and thermal diffusion (Riveros-Iregui et al., 2007;
Phillips et al., 2011; Zhang et al., 2015). In general, many of these studies conclude that diel hysteresis is the result of
physical processes alone. Few studies have evaluated the causes of diel hysteresis in drylands. Currently, it is not clear to
what degree physical and biological processes control hysteresis in drylands.

Drylands are characterized with low productivity. As weak organic C-storage pools (West et al., 1994; Lange, 2003),

drylands are noted for their large contribution of autotrophic production of $CO_2$. The autotrophic component of $R_s$ occurs as a
direct consequence of root respiration, which is firmly coupled (within several hours) to recent photosynthesis (Liu et al.,
2006; Baldocchi et al., 2006; Högberg and Read, 2006; Bahn et al., 2009; Kuzyakov and Gavrichkova, 2010). Consequently,
photosynthesis may govern the level of variation in asynchronicity between $R_s$ and $T_s$ in drylands. In drylands, especially in
desert ecosystems characterized by sandy soils with high soil porosity, the influence of $SWC$ on gas diffusion is likely
nominal. As a rule, most of the available water is used directly in sustaining biological activity in drylands (Noy-Meir, 1973).
Under drought conditions, stomata closure in plants at midday reduces water losses, resulting in a corresponding suppression
of photosynthesis (Jia et al, 2014). Such changes in diel patterns of photosynthesis likely result in modifications of patterns
in $R_s$, leading to hysteresis between $R_s$ and $T_s$. Soil water content likely regulates photosynthesis and, in so doing, causes
hysteresis between $R_s$ and $T_s$ to vary over the growing season.

In this study, we hypothesize that: (1) photosynthesis has a high degree of control in the formation of hysteretic loops

between $R_s$ and $T_s$; and (2) $SWC$ regulates this control and its variation over the growing season. The main objectives of this
research were to: (1) assess biological controls on diel hysteresis between $R_s$ and $T_s$; (2) explore the causes that lead to
variation in seasonal variation in diel hysteresis; and (3) understand $SWC$'s role in influencing hysteresis. To undertake this
work, we measured $R_s$, $SWC$, $T_s$, and photosynthesis in a dominant desert-shrub on a continuous basis for 2013.
**2 Materials and Methods**
**2.1 Site description**
The study was conducted at Yanchi Research Station of Beijing Forestry University, Ningxia, northwest China (37°42'31" N,
107°13'37" E, 1550 m a.s.l). The station is located at the southern edge of the Mu Us desert in the transition between the arid
and semi-arid climatic zones. Based on 51 years of data (1954-2004) from the Meteorological Station at Yanchi, the mean
annual air temperature at the station was 8.1°C and the mean annual total precipitation was 292 mm (ranging between 250 to
350 mm), 63% of which fell in late summer (i.e., July-September; Wang et al., 2014; Jia et al., 2014). Annual potential
evaporation was on average 5.5 kg m$^{-2}$ d$^{-1}$ (Gong et al., 2016). The soil at the research station was of a sandy type, with a
bulk density of 1.6 g cm$^{-3}$. The total soil porosity within 0-2 and 5-25 cm depths was 50% and 38%, respectively. Soil
organic matter, soil nitrogen, and pH were 0.21-2.14 g kg$^{-1}$, 0.08-2.10 g kg$^{-1}$, and 7.76-9.08, respectively (Wang et al., 2014;
Jia et al, 2014). The vegetation was regenerated from aerial seeding applied in 1998 and is currently dominated by a semi-
shrub species cover of *Artemisia ordosica*, averaging about 50-cm tall with a canopy size of about 80 cm × 60 cm (for
additional site description, consult Jia et al. 2014 and Wang et al. 2014, 2015).

## 2.2 Soil respiration and photosynthesis measurement

Two permanent polyvinyl chloride soil collars were initially installed on a small fixed sand dune in March, 2012. Collar dimensions were 20.3 cm in diameter and 10 cm in height, with 7 cm inserted into the soil. One collar was set on bare land with an opaque chamber (LI-8100-104, Nebraska, USA) and the other over an *Artemisia ordosica* plant (~10 cm tall) with a transparent chamber (LI-8100-104C). Soil respiration ($\mu mol\ CO_2\ m^{-2}\ s^{-1}$) was directly estimated from $CO_2$-flux measurements obtained with the opaque-chamber system. Photosynthetic rates ($\mu mol\ CO_2\ m^{-2}\ s^{-1}$) of the selected plants were determined as the difference in $CO_2$ fluxes obtained with the transparent and opaque chambers.

Continuous measurements of $CO_2$ fluxes ($\mu mol\ CO_2\ m^{-2}\ s^{-1}$) were made *in situ* with a Li-8100 $CO_2$-gas analyzer and a LI-8150 multiplexer (LI-COR, Nebraska, USA) connected to each chamber. Instrument maintenance was carried out bi-weekly during the growing season, including removing plant-regrowth in the opaque-chamber installation, and cleaning to avoid blackout conditions associated with the transparent chamber. Measurement time for each chamber was 3 minutes and 15 seconds, including a 30-second pre-purge, 45-second post-purge, and 2-minute measurement period.

## 2.3 Measurements of temperatures, soil water content and other environmental factors

Hourly soil temperature ($T_s$, ºC) and volumetric soil water content ($SWC$, $m^3\ m^{-3}$) at a 10-cm depth were measured simultaneously about 10 cm from the chambers using a LI-8150-203 temperature and $EC_{H2O}$ soil-moisture sensor (LI-COR, Nebraska, USA; see Wang et al., 2014). Other environmental variables were recorded every half hour using sensors mounted on a 6-m tall eddy-covariance tower approximately 800 m from our soil $CO_2$-flux measurement site. Air temperature ($T_a$, ºC) was measured with a thermohygrometer (HMP155A, Vaisala, Finland). Soil-surface temperature ($T_{surf}$, ºC) was measured with an infrared-emission sensor (Model SI-111, Campbell Scientific Inc., USA). Incident photosynthetically active radiation ($PAR$) was measured with a light-quantum sensor (PAR-LITE, Kipp and Zonen, the Netherlands) and precipitation ($PPT$, mm), with three tipping-bucket rain gages (Model TE525MM, Campbell Scientific Inc., USA) placed 50 m from the tower (see Jia et al., 2014).

## 2.4 Data processing and statistical analysis

In this study, $CO_2$-flux measurements were screened by means of limit checking, i.e., hourly $CO_2$-flux data < -30 or > 15 $\mu mol\ CO_2\ m^{-2}\ s^{-1}$ were considered to be anomalous as a result of, for instance, gas leakage or plant damage by insects, and removed from the dataset (Wang et al., 2014, 2015). After limit checking, hourly $CO_2$ fluxes greater than three times the standard deviation from the calculated mean of 5 days' worth of flux data were likewise removed. Quality control and instrument failure together resulted in 5% loss of hourly fluxes for all chambers, 4% for temperatures, and 8% for $SWC$ (Fig. 1). Differences in mean annual $T_s$ and $SWC$ between the two chambers were 0.01 ºC and 0.003 $m^3\ m^{-3}$, respectively.

The $Q_{10}$-function (e.g., Eq. 1) was used here to describe the response of $R_s$ to temperature. Earlier studies have shown

strong correlation between basal rate of $R_s$ and photosynthesis (Irvine et al., 2005; Sampson et al., 2007). Response of $R_s$ to
changes in photosynthesis was, in turn, characterized as a linear function (Eq. 2). Interaction between photosynthesis and
temperature on $R_s$ was conveyed through Eq. 3. The instantaneous relative importance ($RI$) of photosynthesis and
temperature on $R_s$ over the growing season was calculated with a correlation-based ratio (see Eq. 4). The importance of
photosynthesis on $R_s$ increases with a corresponding increase in $RI$:
$R_s = R_{10} \times Q_{10}^{(T-10)/10}$           (1)
$R_s = a \times P + b$           (2)
$R_s = (a \times P + b) \times c^{(T-10)/10}$           (3)
$RI = \dfrac{\rho_p}{\rho_t}$           (4)
where $R_{10}$ is the respiration at 10°C, $Q_{10}$ is the temperature sensitivity of respiration, $T$ is temperature, $P$ is photosynthesis
($\mu$mol $CO_2$ m$^{-2}$ s$^{-1}$), $a$, $b$, and c are regression coefficients, and $\rho_p$ and $\rho_t$ are the correlation coefficients between
photosynthesis and $R_s$ and temperature and $R_s$, respectively.

Pearson correlation analysis was used to calculate the correlation coefficient between temperature or photosynthesis and

$R_s$. Cross-correlation analysis was used to estimate hysteresis in the relationship between temperature and $R_s$ and
photosynthesis- and $R_s$. We used root mean squared error ($RMSE$) and the coefficient of determination ($R^2$) as criteria in
evaluating function performance. To evaluate seasonal variation in diel hysteresis, the mean monthly daily cycles of $R_s$, $T_a$,
$T_{surf}$, $T_s$, and photosynthesis were generated by averaging their hourly means at a given hour over a particular month (Table
1). Exponential and linear regression was used to evaluate the influence of $SWC$ on the control of photosynthesis on
temperature-$R_s$ hysteresis. Likewise, influences of $SWC$ on diel hysteresis was examined during a wet month with high
rainfall and adequate $SWC$ (July, $PPT$ = 117.9 mm) and a dry month with low rainfall and inadequate $SWC$ (August, $PPT$ =
10.9 mm; Wang et al., 2014). In order to evaluate the influence of photosynthesis on diel hysteresis in the temperature-$R_s$
relationship, we compared the time lag (in hours) between measured and modeled $R_s$ by means of Eq.'s 1 through 3 with a
one-day moving window and a one-day time step over the growing season (April to October). Modeled $R_s$ was calculated
using the fitted parameters of each function and the measured hourly $T_{surf}$ and photosynthesis for each day. All statistical
analyses were performed in MATLAB, with a significance level of 0.05 (R2010b, Mathworks Inc., Natick, MA, USA).
**3 Results**
**3.1 Diel patterns of soil respiration, photosynthesis, and environmental factors**
Incident photosynthetically active radiation, $T_a$, $T_{surf}$, and $T_s$ exhibited distinctive daily patterns over the year (Fig. 1a-d),
peaking at ~12:00 PM (Local Time, LT), ~16:00 PM, ~14:00 PM, and ~17:00 PM, respectively (Fig. 1a-d). Unlike the
environmental factors, daily patterns in $R_s$ remained constant over the non-growing part of the year, peaking at 11:00 AM-
13:00 PM, and highly variable during the growing season of the year (April to October), peaking between 10:00 AM-16:00
PM (Fig. 1f). Similar to $R_s$ during the growing season, diel patterns of photosynthesis were also highly variable, peaking
between 10:00 AM-16:00 PM (Fig. 1e).
Diel patterns of monthly mean $R_s$ were similar to those of $T_{surf}$ during the wet month and similar to those of
photosynthesis during the dry month (Fig. 2g, h). During the wet month (July), monthly mean diel $R_s$ was out of phase with
photosynthesis, but in phase with $T_{surf}$ (Fig. 2g). Soil respiration peaked at 16:00 PM, exhibiting similar timing to $T_{surf}$ (i.e.,
15:00 PM), but four hours later than photosynthesis (peaking at 12:00 PM; Fig. 2g). During the dry month (August), diel $R_s$
was generally in phase with photosynthesis, but out of phase with $T_{surf}$ (Fig. 2h). Both photosynthesis and $R_s$ plateaued
between 10:00 AM-16:00 PM, whereas $T_{surf}$ peaked at 15:00 PM (Fig. 2h).

**3.2 Control of photosynthesis and temperature on diel soil respiration**

Among temperatures at the three levels, $T_{surf}$ correlated the strongest with $R_s$, due to the high $R^2$'s with monthly mean diel $R_s$
(Table 1). Over the growing season, monthly mean diel $R_s$ correlated fairly well with photosynthesis (Table 1). The response
of $R_s$ to temperature and photosynthesis was shown to be affected by $SWC$ (Table 2, Fig. 3). During the wet month, $T_{surf}$
alone explained 97% of the variation in diel $R_s$ (via Eq. 1), whereas photosynthesis explained 67% of the variation (Table 2,
Fig. 3a). However, during the dry month, photosynthesis explained 88% of the variation in diel $R_s$ (via Eq. 2), whereas $T_{surf}$
explained 76% of the variation (Fig. 3b, Table 2). Irrespective of dry or wet periods, $T_{surf}$ and photosynthesis together
explained over 90% of the diel variation in $R_s$ (via Eq. 3; see Fig 3 and Table 2). On the whole, $RI$ varied as a function of
$SWC$, decreasing whenever $SWC$ increased (Fig. 4).

**3.3 Effects of soil water content and photosynthesis on diel hysteresis in temperature-$R_s$ relationship**

During the wet month, hysteresis was not observed to occur in the monthly mean $T_{surf}$-$R_s$ relationship, whereas two-hour lags
were found to occur in the photosynthesis-$R_s$ relationship (Table 1; Fig. 3a). During the dry month, the opposite was
observed, where one-hour lags were found to occur in the $T_{surf}$-$R_s$ relationship (Table 1, Fig. 3b). Over the growing season,
$T_{surf}$ lagged behind $R_s$ by about 0-4 hours (Fig. 5b), and $R_s$ lagged behind photosynthesis by about the same amount (Fig. 5c).
This led to time lags between measured and modeled $R_s$ regardless of the variable, $T_{surf}$ or photosynthesis, resulting in about
26% of the days of the growing season (accounting for 184 days, in total) having no time lag (Fig. 5e, f). However, taking
into account both $T_{surf}$ and photosynthesis as input variables in the definition of $R_s$ (via Eq. 3), time lags between measured
and modeled $R_s$ were mostly eliminated (Fig. 5a, d), with 84% of the days of the growing season displaying no time lag.

Diel hysteresis in both relationships (i.e., $T_{surf}$-$R_s$ and photosynthesis-$R_s$) was shown to be affected by $SWC$ (Fig. 6).

Over the growing season, diel hysteresis between $R_s$ and $T_{surf}$ was linearly related to $SWC$ in a downward manner, when $SWC$
$< 0.08$ m$^3$ m$^{-3}$ (ratio of $SWC$ to soil porosity = 0.26; Fig. 6a). Hysteresis was not evident, when $SWC > 0.08$ m$^3$ m$^{-3}$ (Fig. 6a).
In contrast, diel hysteresis between $R_s$ and photosynthesis was linearly related to $SWC$ in an upward manner, when $SWC <$
$0.08$ m$^3$ m$^{-3}$ (Fig. 6b), but ceased to be related, when $SWC > 0.08$ m$^3$ m$^{-3}$ (Fig. 6b).
**4 Discussion**
**4.1 Degree of control of photosynthesis on diel hysteresis**
In our study, we found that the diurnal pattern in temperature ($T_a$, $T_{surf}$, and $T_s$) lagged behind $R_s$ by 0-4 hours, which resulted
in a counterclockwise loop in the relationship between $R_s$ and temperature. Although the magnitude of hysteresis between $R_s$
and temperature differed among the three temperature measurements, their seasonal variation was generally uniform. Among
the temperature measurements, $T_{surf}$ was more closely related to diel $R_s$, resulting in weaker hysteresis. Magnitude of
hysteresis between $R_s$ and temperature was comparable to those in other plant systems, e.g., 3.5-5 h in a boreal aspen stand
(Gaumont-Guay et al., 2006) and 0-5 h in a Chinese pine plantation (Jia et al., 2013). However, the direction of hysteresis
was unlike that reported by Phillips et al. (2011), who had reported $R_s$ lagging behind soil temperature.

In general, transfer of heat (downward) and gases (upward) through the soil complex by simple diffusion would take

time to occur. Increased $SWC$ would serve to impede this transfer (Millington and Quirk, 1961). If physical processes alone
controlled hysteresis, you would expect $R_s$ to lag behind $T_{surf}$ and hysteresis to increase with increasing $SWC$. However, such
rationalization is not supported by our observations, which show $T_{surf}$ to lag behind $R_s$ and hysteresis to decrease with
increasing $SWC$. As a result, physical processes alone cannot account for the observed patterns in hysteresis between $R_s$ and
temperature. Combining photosynthesis and $T_{surf}$ as explanatory variables of $R_s$ (via Eq. 3), we found 84% of the days over
the growing season had no observable lag between measured- and modeled-$R_s$, relative to 27% of the days when $T_{surf}$ alone
was used (associated with to Eq. 2), suggesting that photosynthesis has an important role in governing hysteresis in desert
shrubland. Along with other studies, including those of Tang et al. (2005), Vargas and Allen (2008), Carbone et al. (2008),
Kuzyakov and Gavrichkova (2010), and Wang et al. (2014), our findings provide increasing evidence of the role of
photosynthesis in regulating diel hysteresis between $R_s$ and temperature.
**4.2 Photosynthesis control of soil respiration and diel hysteresis**
The 0-4 h lag between $R_s$ and photosynthesis observed are consistent with those observed in earlier studies, e.g., 0-4 h lag
between ecosystem-level photosynthesis and $R_s$ in a coastal wetland ecosystem (Han et al., 2014) and 0-3 h lag between
plant photosynthesis and $R_s$ in a steppe ecosystem (Yan et al., 2011). Short time lags suggest rapid response between recent
photosynthesis and $R_s$ (Kuzyakov and Gavrichova, 2010). This response is significantly faster than suggested in earlier
studies, when approached from an isotopic or canopy/soil flux-based methodology (Howarth et al., 1994; Mikan et al., 2000;
Jonson et al., 2002; Högberg et al., 2008; Kuzyakov and Gavrichova, 2010; Mencuccini and Hölttä, 2010; Kayler et al., 2010;
Han et al., 2014).
According to the "goodness-of-fit" of Eq. 3 to the field data, the time lag between diel photosynthesis and $R_s$ was likely
caused by variations in temperature, regardless of $SWC$. Photosynthesis provide substrates to roots and rhizosphere microbes
(Tang et al., 2005; Kuzyakov and Gavrichkova, 2010; Vargas et al., 2011; Han et al., 2014). Temperature directly drives
enzymatic kinetics of respiratory metabolism in organisms (Van't Hoff, 1898; Lloyd and Taylor, 1994). Photosynthesis is
directly driven by radiation (specifically, photosynthetically active radiation). Temperature is also driven by radiation, but
through heating of the surface and subsequent air and soil layers. Thus, diel patterns in temperature continuously lagged
behind those of photosynthesis by a few hours (as indicated in Fig. 2). The interactions between photosynthesis and
temperature led $R_s$ to lag behind photosynthesis, but temperature lagged behind $R_s$ (Fig. 2). This sequence of events may
explain the difference in the direction of hysteresis observed here, in contrast to that reported in Phillips et al. (2011). Such
explanation is different from the explanations for forest ecosystems, where the transport of photosynthates and influence of
turgor and osmotic pressure may be responsible for the specific coupling observed between current photosynthesis and $R_s$
(Steinmann et al., 2004; Högberg et al., 2008; Hölttä et al., 2006, 2009; Mencuccini and Hölttä, 2010). Variations in
coupling dynamics may occur because of differences in vegetation height among ecosystems (Kuzyakov and Gavrichova,
2010; Mencuccini and Hölttä, 2010). Unlike forest ecosystems, low-statured vegetation in shrub systems (~0.5 m), may elicit
a few minutes of delay in the transportation of photosynthates and influence of turgor and osmotic pressure (Kuzyakov and
Gavrichkova, 2010). Such small time lags cannot be easily identified in hourly measurements, resulting in an apparent
temperature-dominated control of photosynthesis and $R_s$.

### 4.3 Influences of soil water content on seasonal variation in diel hysteresis

Diel $R_s$ varied consistently with $T_{surf}$, with no observable signs of hysteresis, when $SWC > 0.08$ m$^3$ m$^{-3}$. However, as $SWC$
decreased from this value, diel $R_s$ varied more closely with photosynthesis, leading to increased diel hysteresis between $R_s$
and $T_{surf}$. These results suggest that $SWC$ played a more important role in regulating the relative control of photosynthesis
and temperature on diel $R_s$ over the growing season, supporting our second hypothesis.
A possible explanation for $SWC$ regulating hysteresis might be associated with changes in substrate supply. During the
wet period with $SWC > 0.08$ m$^3$ m$^{-3}$, increases in $SWC$ ameliorates diffusion of soil C substrates and its access to soil
microbes (Curiel Yuste et al., 2003; Jarvis et al., 2007). Amount of substrate to roots and rhizosphere microbes is also
expected to be high as a result of high current photosynthesis (Baldocchi et al., 2006). As a result, diel $R_s$ is not limited by C
substrates provided by current photosynthesis and soil organic matter. Consequences of diel $R_s$ may vary repeatedly in
synchrony with diel temperature, with no indication of hysteresis when $SWC > 0.08$ m$^3$ m$^{-3}$ (Fig. 6a). By contrast, during dry
and hot phases, with $SWC < 0.08$ m$^3$ m$^{-3}$, inadequate soil water limits diffusion of soil C substrates and its access to soil
microbes (Jassal et al., 2008) and also suppresses photosynthesis (supported by Fig. 2g, h). As a result, $R_s$ may be limited by
C substrates under dry conditions. It has been reported current photosynthesis can account for about 65-70% of total $R_s$ over
the growing season (Ekblad and Högberg et al., 2001; Högberg et al., 2001). Thus, diel $R_s$ may vary more closely to
photosynthesis during dry and hot phases over the growing season (Fig. 2h), resulting in increased hysteresis with decreasing
*SWC* below 0.08 m$^3$ m$^{-3}$ (Fig. 6b).
The 0.08 m$^3$ m$^{-3}$ *SWC* threshold of this study was consistent with an earlier study by Wang et al. (2014) that reported
that seasonal $R_s$ decoupled from soil temperature as *SWC* fell below 0.08 m$^3$ m$^{-3}$. Earlier studies have reported similar
responses of $R_s$ to temperature (Palmroth et al., 2005; Jassal et al., 2008). For example, $R_s$ in an 18-year-old temperate
Douglas-fir stand decoupled from $T_s$ when *SWC* fell below 0.11 m$^3$ m$^{-3}$. Our results suggest that the decoupling of $R_s$ from
temperature for low *SWC* was due to a shift in control from temperature to photosynthesis. Our work provides urgently
needed new knowledge concerning causes/mechanisms involved in defining variation in diel hysteresis in desert shrubland.
Based on our work, we suggest that photosynthesis should be considered in simulations of diel $R_s$ in drylands, especially
when *SWC* falls below 0.08 m$^3$ m$^{-3}$.

## 254  5 Conclusions

Soil water content regulated the relative control between photosynthesis and temperature on diel $R_s$ by changing the relative
contribution of autotrophic and heterotrophic respiration to total $R_s$, causing seasonal variation in diel hysteresis between $R_s$
and temperature. Hysteresis was not observed between $R_s$ and $T_{surf}$, when *SWC* > 0.08 m$^3$ m$^{-3}$, but the lag-hours increased as
*SWC* decreased below this *SWC* threshold. Incorporating photosynthesis into $R_s$-temperature-based models reduces diel
hysteresis and increases the overall level of goodness-of-fit. Our findings highlight the importance of biological mechanisms
in diel hysteresis between $R_s$ and temperature and the importance of *SWC* in plant photosynthesis-soil respiration dynamics
in dryland ecosystems.

*Acknowledgement.* We acknowledge the grants obtained from National Natural Science Foundation of China (NSFC)
(31670710 and 31361130340), the Fundamental Research Funds for the Central Universities (BLYJ201601), and the
Finnish-Chinese research collaboration project EXTREME (2013-2016), between Beijing Forestry University and University
of Eastern Finland (EXTREME proj. 14921 funded by Academy of Finland). Also the U.S.-China Carbon Consortium
(USCCC) supported this work by way of helpful discussions and exchange of ideas. We acknowledge Dr. Paul Stoy,
Associate Editor, and anonymous reviewers for their valuable comments and suggestions on this manuscript. We also
acknowledge Huishu Shi, Yuming Zhang, Wei Feng, Yajuan Wu, Peng Liu, Qiang Yang, and Mingyan Zhang for their
assistance with the field measurements and instrumentation maintenance.

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

**Table 1.** Analysis of mean monthly diel cycles of soil respiration ($R_s$), air temperature ($T_a$), soil-surface temperature ($T_{surf}$), soil
temperature at a 10-cm depth ($T_s$), and photosynthesis ($P$) in a dominant desert-shrub ecosystem, including correlation coefficients and
time lags times in $R_s$ vs. $T_a$, $T_{surf}$, $T_s$, and $P$ cycles. Statistically significant Pearson's correlation coefficients ($r$; $p < 0.05$) are denoted in
bold.

| | | Jan | Feb | Mar | Apr | May | Jun | Jul | Aug | Sep | Oct | Nov | Dec |
|---|---|---|---|---|---|---|---|---|---|---|---|---|---|
| $R_s$-$T_a$ | Lag | 2 | 4 | 3 | 3 | 1 | 1 | 1 | 2 | 1 | 1 | 1 | 1 |
| | $r$ | **0.64** | 0.25 | **0.49** | **0.46** | **0.85** | **0.85** | **0.93** | **0.76** | **0.94** | **0.89** | **0.78** | **0.77** |
| $R_s$-$T_{surf}$ | Lag | 1 | 2 | 2 | 2 | 0 | 0 | 0 | 1 | 0 | 0 | 1 | 1 |
| | $r$ | **0.82** | **0.57** | **0.75** | **0.72** | **0.96** | **0.96** | **0.98** | **0.87** | **0.98** | **0.97** | **0.89** | **0.87** |
| $R_s$-$T_s$ | Lag | 4 | 5 | 5 | 5 | 3 | 3 | 2 | 4 | 2 | 2 | 4 | 4 |
| | $r$ | -0.06 | -0.31 | -0.06 | -0.07 | **0.54** | **0.58** | **0.80** | 0.31 | **0.77** | **0.65** | 0.23 | 0.12 |
| $R_s$-$P$ | Lag | | | | | -1 | -1 | -2 | 0 | -1 | -1 | | |
| | $r$ | | | | | **0.84** | **0.83** | **0.82** | **0.94** | **0.86** | **0.88** | | |

**Table 2.** Regressions based on the $Q_{10}$, linear, and $Q_{10}$-linear functions of soil respiration ($R_s$) for both a wet (July) and dry month (August)
in 2013. Variables $T_{surf}$ (°C) refers to the soil-surface temperature; $P$ photosynthesis in the dominant shrub layer; $R^2$ the coefficient of
determination; and $RMSE$ the root mean squared error.

| | Model | Wet month: July | Dry month: August |
|---|---|---|---|
| $R_s$-$T$ | $Q_{10}$ | $R_s = 1.13 \times 1.4^{\frac{T_{surf}-10}{10}}$ | $R_s = 1.12 \times 1.1^{\frac{T_{surf}-10}{10}}$ |
| | | $R^2 = 0.97$ | $R^2 = 0.76$ |
| | | $RMSE = 0.0521$ | $RMSE = 0.0796$ |
| $R_s$-$P$ | Linear | $R_s = 0.03 \times P + 1.61$ | $R_s = 0.04 \times P + 1.29$ |
| | | $R^2 = 0.67$ | $R^2 = 0.88$ |
| | | $RMSE = 0.1889$ | $RMSE = 0.05752$ |
| $R_s$-$P$-$T$ | Linear$\times Q_{10}$ | $R_s = (0.002 \times P + 1.16) \times 1.38^{\frac{T_{surf}-10}{10}}$ | $R_s = (0.024 \times P + 1.20) \times 1.08^{\frac{T_{surf}-10}{10}}$ |
| | | $R^2 = 0.98$ | $R^2 = 0.94$ |
| | | $RMSE = 0.0491$ | $RMSE = 0.0408$ |


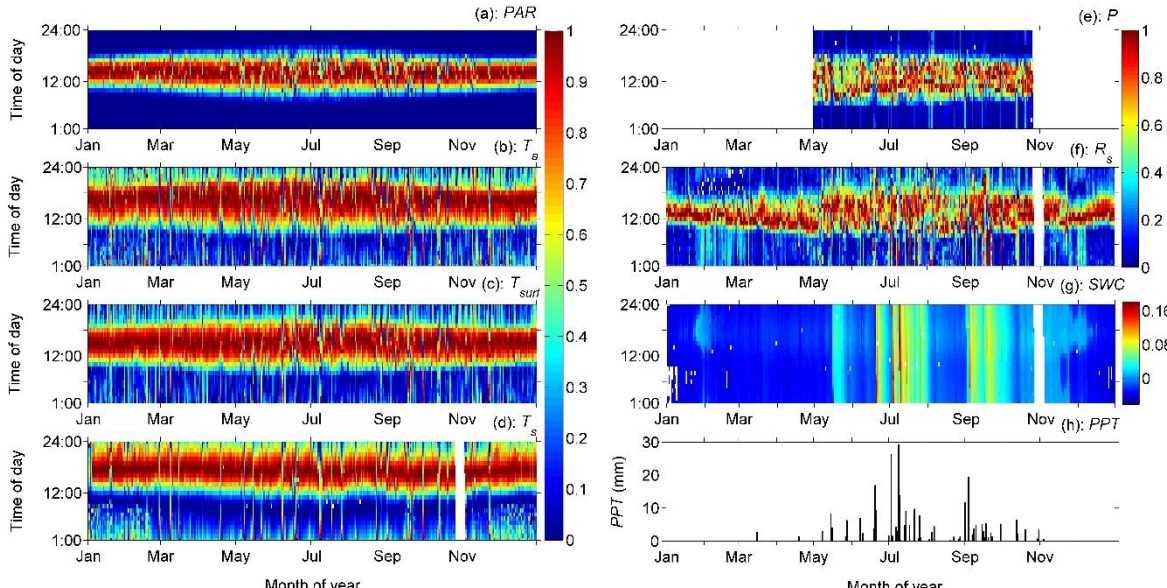


**Figure 1.** Seasonal variation in incident photosynthetically active radiation (*PAR*), temperature [i.e., air ($T_a$), soil-surface ($T_{surf}$), and soil
temperatures ($T_s$)], photosynthesis (*P*), and soil respiration ($R_s$) at an *Artemisia ordosica*-dominated site, and seasonal variation in soil
water content (*SWC*) and precipitation (*PPT*) for 2013. Hourly *PAR*, $T_a$, $T_{surf}$, $T_s$, $R_s$, and *P* are normalized against all values for each day.
Each hourly value (*y*-axis) for each day (*x*-axis) is shown as a value of 1 through 0; 1 denotes the peak value for a given day and 0, the
daily minimum value.

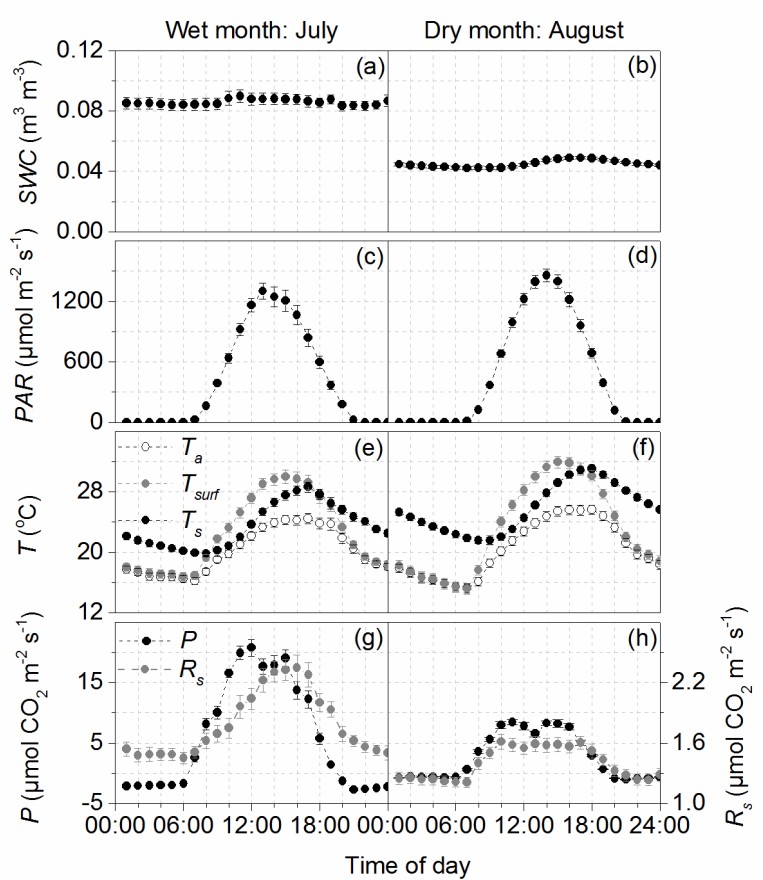


**Figure 2.** Mean monthly diel cycle of soil water content (*SWC*), incident photosynthetically active radiation (*PAR*), temperature [i.e., air

($T_a$), soil-surface($T_{surf}$), and soil temperatures ($T_s$)], soil respiration ($R_s$), and photosynthesis (*P*) at an *Artemisia ordosica*-dominated site

during a wet and dry month. Each point is the monthly mean for a particular time of day. Bars represent standard errors.


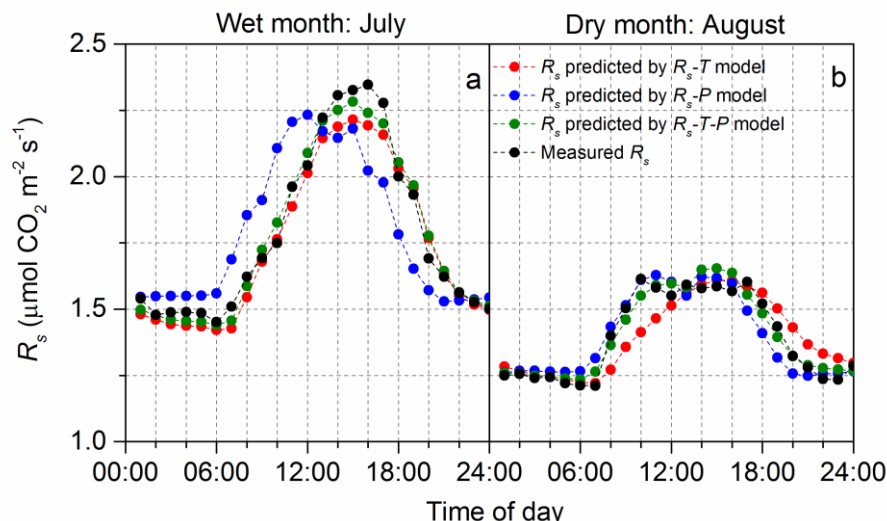

**Figure 3.** Diel variation of measured soil respiration ($R_s$) and modeled $R_s$ by using temperature and photosynthesis as input variables in the
calculation of $R_s$ for both a wet and dry month (i.e., July and August, respectively); $R_s$-$T$ function (Eq. 1), $R_s$-$P$ function (Eq .2), and $R_s$-$T$-$P$
function (Eq. 3).

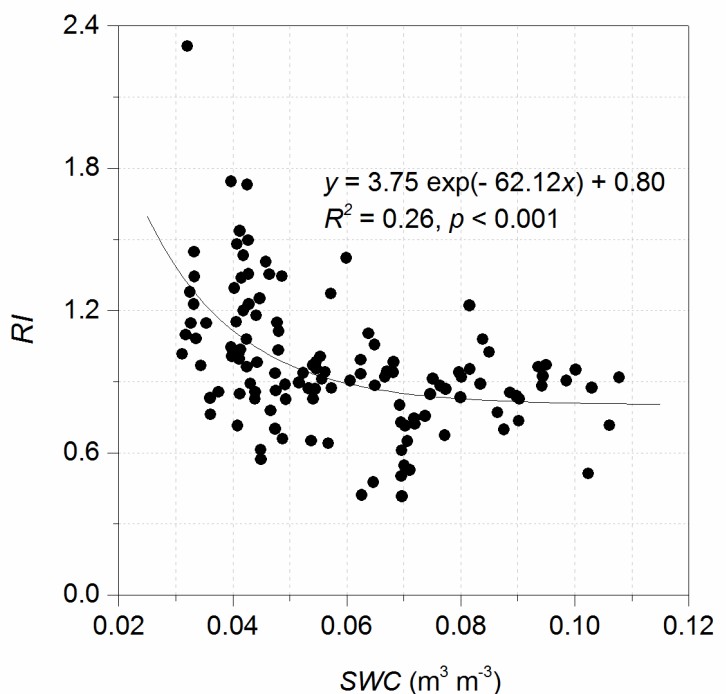

$y = 3.75 \exp(-62.12x) + 0.80$
$R^2 = 0.26$, $p < 0.001$

**Figure 4.** Relationship between soil water content (*SWC*) and the relative importance (*RI*) of soil-surface temperature and photosynthesis
at an *Artemisia ordosica*-dominated site as a function of soil respiration ($R_s$).

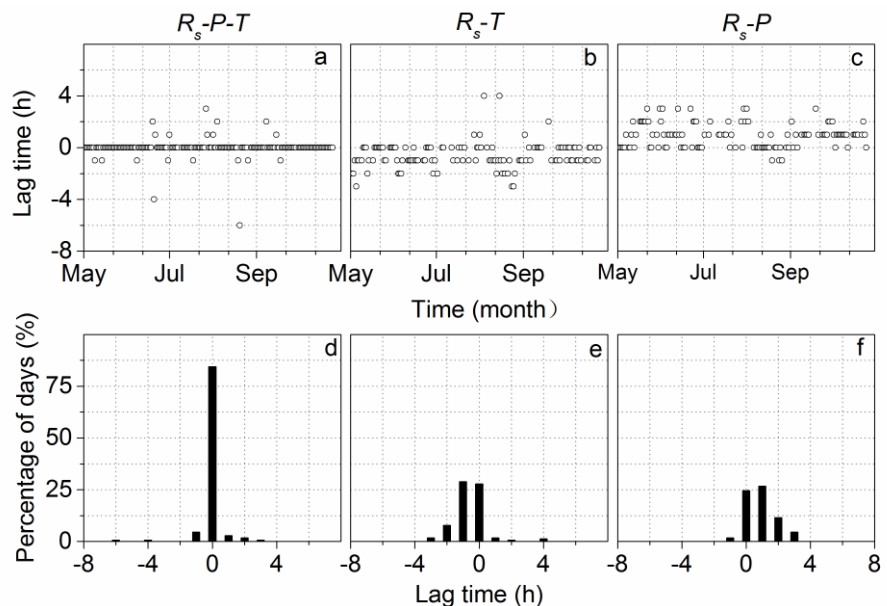


**Figure 5.** Time lags between measured and modeled soil respiration by means of soil-surface temperature and photosynthesis over the
growing season; $R_s$-$T$ function (Eq. 1), $R_s$-$P$ function (Eq. 2), and $R_s$-$P$-$T$ function (Eq. 3).

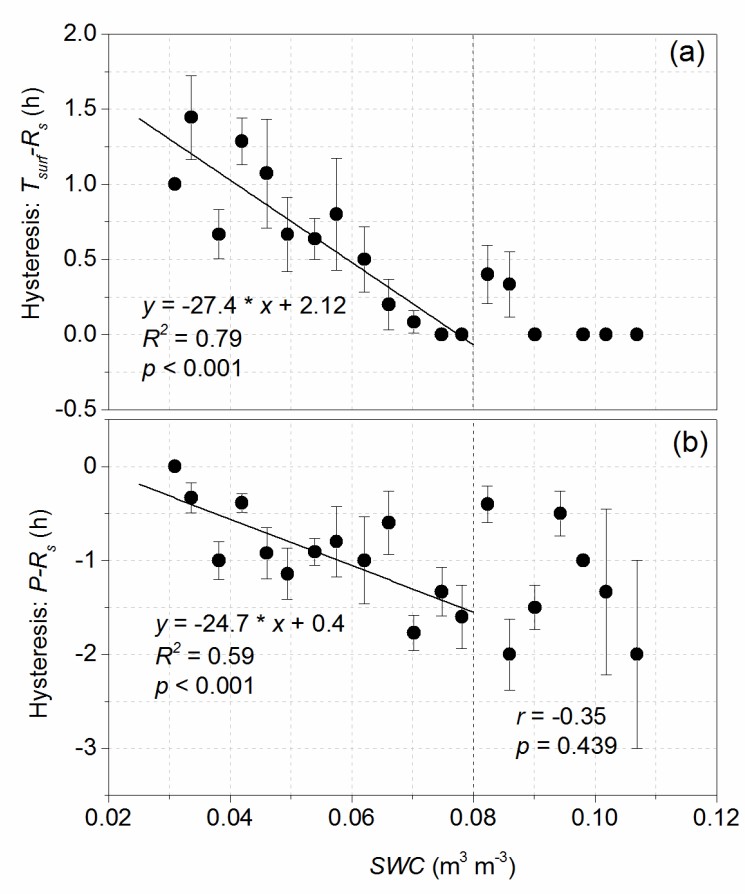


**Figure 6.** Time lags between soil respiration ($R_s$) and soil-surface temperature ($T_{surf}$), $R_s$, and photosynthesis at an *Artemisia ordosica*-

dominated site with respect to soil water content ($SWC$). Time lags were bin-averaged using $SWC$-intervals of 0.004 $m^3$ $m^{-3}$