# Peer review of "Soil water regulates the control of photosynthesis on diel hysteresis"

_Biogeosciences, 2016_

## Referee Comment (RC1) · Anonymous Referee #1 · 16 Nov 2016

Although the topic of hysteresis of Rs and Ts has been discussed in other ecosystems, and the method and statistical method is regular, the impact of SWC on diel hysteresis is well explained in this manuscript. The data and figures are well organized. The author has provided us a relatively complete story of how low SWC affect Rs by restraining biological process, like photosynthesis. The elliptical loop of Rs and Ts diel hysteresis gives a clear explanation. However, this thesis did not explain the mechanism clearly. The mechanism of SWC regulate diel hysteresis between Rs and Tsurf needs strong evidence to explicate other than just mentioning other people's deduction that it is due to relative contribution of autotrophic and heterotrophic respiration to total Rs. This manuscript needs carefully organize the proof of how and why respiration

component affects Rs in dry land.

---

## Referee Comment (RC2) · Anonymous Referee #2 · 26 Nov 2016

In this manuscript, by introducing a threshold value of soil water content (SWC), the dominating factor of diel variation of soil respiration is identified as photosynthesis or temperature, corresponding to autotrophic or heterotrophic respiration. Generally, the MS is well written and the topic is interesting, however, it would be better to substitute the ratio of SWC to soil porosity for SWC. As the author mentioned in "4.1 Physical- vs. biological-processes in the control of diel hysteresis", within-soil gas transport is influenced not only by SWC, but also by soil porosity. So with the ratio of SWC to soil porosity, the results and conclusion of this MS will be more comparable and universal, but not so local.
* * *

---

## Author Comment (AC1) · 5 Dec 2016

Response to referee's comments

We thank referee and greatly appreciate the thoughtful and constructive comments. We have fully considered the referee's comments in the revision and improved the manuscript. Answers to referee's questions are italicized and in blue color.

General comments:

Although the topic of hysteresis of Rs and Ts has been discussed in other ecosystems, and the method and statistical method is regular, the impact of SWC on diel hysteresis is well explained in this manuscript. The data and figures are well organized. The

author has provided us a relatively complete story of how low SWC affect Rs by restraining biological process, like photosynthesis. The elliptical loop of Rs and Ts diel hysteresis gives a clear explanation. However, this thesis did not explain the mechanism clearly. The mechanism of SWC regulate diel hysteresis between Rs and Tsurf needs strong evidence to explicate other than just mentioning other people's deduction that it is due to relative contribution of autotrophic and heterotrophic respiration to total Rs. This manuscript needs carefully organize the proof of how and why respiration component affects Rs in dry land.

Answers:

In our study, we didn't have available measurements on autotrophic and heterotrophic respiration, making possible to directly determine the relative changes between the two components of soil respiration (Rs) experimentally. However, our comparatively statistical analyses on the patterns of diurnal variation in soil respiration (Rs) and soil surface temperature (Tsurf) between the wet and dry month (Figure 2), and the relationship between relative importance of temperature and photosynthesis (RI) and soil water content (SWC) (Figure 4) were significantly enough to indirectly understand the relative changes between the two components.

In desert shrub land, soil organic matters are thin and are mainly concentrated on the soil surface (Ciais et al., 2011; Thomas, 2012; Gao et al., 2014), thus heterotrophic respiration responds primarily to Tsurf, and thus root and rhizosphere respiration is firmly associated with photosynthesis. That is to say that autotrophic respiration responds primarily to photosynthesis.

Under low SWC condition in summer, soil surface temperature was normally higher, decomposition processes of organic matters was largely depressed, whereas roots system spread deeply into soil, thus plant may use deep soil water to maintain metabolism. Thus, autotrophic respiration dominantly contributes to Rs under low SWC, resulting in Rs being correlated more to photosynthesis than Tsurf (line 10-11 on P6; Figure 2),

leading to high RI (line 11-12 on P6; Figure 4). In contrast, under high SWC condition after rainfall events, the decomposition process was largely accelerated, as we commonly reported large $CO_2$ fluxes purse after rainfall events. Thus relative contribution of heterotrophic respiration largely increased after rainfall events, resulting in Rs being correlated more to Tsurf than photosynthesis (line 8-9 on P6; Figure 2), and leading to low RI (line 11-12 on P6; Figure 4).

Base on above findings, we assumed that the regulation of SWC on diel hysteresis between Rs and Tsurf is affected by the changes in relative contribution between autotrophic and heterotrophic respiration to total Rs. The measurements of autotrophic and heterotrophic respiration are considered in our upcoming studies, and may then confirm our assumption experimentally.

We also added three references, see below, to the manuscript in line 27 on P7 and also into references part (see attach supplement file).

References:

Ciais, P., Bombelli, A., Williams, M., Piao S. L., Chave, J., Ryan, C. M., Henry, M., Brender, P. and Valentini, R.: The carbon balance of Africa: synthesis of recent research studies, Philosophical Transactions of the Royal Society of London A: Mathematical, Physical and Engineering Sciences, 369 (1943), 2038-2057, 2011.

Thomas, A. D.: Impact of grazing intensity on seasonal variations of soil organic carbon and soil CO2 efflux in two semi-arid grasslands in southern Botswana, Philosophical Transactions of the Royal Society of London B: Biological Sciences, 367(1606), 3076-3086, 2012.

Gao, G. L., Ding, G. D. Zhao, Y. Y., Wu, B., Zhang Y. Q., Qin S. G., Bao, Y. F., Yu, M. H. and Liu, Y. D.: Fractal approach to estimating changes in soil properties following the establishment of Caragana Korshinskii shelterbelts in Ningxia, NW China, Ecological indicators, 43, 236-243, 2014.

Please also note the supplement to this comment:
http://www.biogeosciences-discuss.net/bg-2016-438/bg-2016-438-AC1-supplement.zip

───────────────────────────

---

## Author Comment (AC2) · 5 Dec 2016

Response to referee's comments

We thank referee and greatly appreciate the thoughtful and constructive comments and helpful suggestions. We have fully considered the referee's comments in the revision and improved the manuscript accordingly.

General comments:

In this manuscript, by introducing a threshold value of soil water content (SWC), the dominating factor of diel variation of soil respiration is identified as photosynthesis or temperature, corresponding to autotrophic or heterotrophic respiration. Generally, the

[Figure]

MS is well written and the topic is interesting, however, it would be better to substitute the ratio of SWC to soil porosity for SWC. As the author mentioned in "4.1 Physicalvs. biological-processes in the control of diel hysteresis", within-soil gas transport is influenced not only by SWC, but also by soil porosity. So with the ratio of SWC to soil porosity, the results and conclusion of this MS will be more comparable and universal, but not so local.

Answer:

We agree with the above-described comments. Following the referee suggestion, we added the text '(ratio of SWC to soil porosity 0.26)' behind the threshold (0.08 m3 m-3) in Introduction (line 7 on P1), Results (line 19 on P6), Discussion (line 19 on P7) and Conclusions (line 18 on P8) paragraphs in the manuscript. However, we still wanted to show also the SWC threshold there. (see attached supplement file)

Please also note the supplement to this comment:
http://www.biogeosciences-discuss.net/bg-2016-438/bg-2016-438-AC2-supplement.zip

---

## Author Response (AR1)

**Responses to editor's comments**

We thank editor and greatly appreciate the thoughtful and constructive comments and helpful suggestions. We have fully considered the editor's comments in the revision and improved the manuscript accordingly. Responses to editor's questions are in blue.

**Associate Editor Decision: Reconsider after major revisions** (02 Feb 2017) by Paul Stoy

**Comments to the Author:**
**Generally comments:**
My comments refer to 'Track change_comment.pdf' found in the online material. Unlike the referees, I found the analysis to be lacking in a number of regards that require clarification and reanalysis. The title is over-generalized; this result is only for desert systems and really only with two respiration chambers. Whereas this is an important measurement flaw that needs to be addressed, I'm not of the opinion that it itself makes the manuscript unpublishable given that eddy covariance observations (likewise a n of 1) are so frequently used to measure ecosystem carbon flux. I don't doubt the basic findings, but the lack of any mechanistic treatment of transport of photosynthate and carbon dioxide within the phloem and soil (respectively) weaken the arguments and compromise their generality.
As a whole, substantial improvements must be made before this manuscript can be reconsidered for publication. Arguments must be strengthened throughout, mechanistic understanding must be incorporated into empirical findings, and the discussion section should not merely be a modified version of the introduction that briefly incorporates the present findings into the existing literature and overextrapolates the findings of the present study across desert ecosystems.

**Response:** We accept and changed the title to "Soil water regulates the control of photosynthesis on diel hysteresis between soil respiration and temperature in a desert shrub land".

The main purpose of this study was to investigate the seasonal variation in diel hysteresis and the causes of such variation. Using more chambers definitely would have increased the accuracy of our $R_s$ measurements, but this has little to do with characterizing time trends in $R_s$. As our field measurements show, the magnitude of $R_s$ differed among chambers, but the time trends were essentially the same. Although only two chambers were used, we make continuous measurements over an entire year. We think the long series of continuous data from two chambers enable us to study temporal variation in the various variables and, thus, allows us to say something about hysteresis.

Indeed, we believe that the rate of photosynthate transport, as well as plant height, are important factors in determining the level of variation in diel hysteresis. Plant height is shown by other researchers to correlate fairly well with diel hysteresis

in various ecosystems (Kuzyakov and Gavrichkova, 2010). Height growth in *Artemisia ordosica* is generally slow, at about 3-4 cm yr$^{-1}$ for our site. This suggests that plant height potentially played a minor role in affecting seasonal variation in diel hysteresis in our study. Moreover, phloem transport rates vary from 0.2 to 2 m h$^{-1}$ (Kuzyakov and Gavrichkova, 2010), suggesting that maximum delay in hysteresis by photosynthate transport alone should be about 30 min. This is clearly incompatible with our own findings, where lag times could be as large as 5 h (Table 1). To make our arguments more generalized, we added this discussion in Section 4.2, See Page 7, line 19.

Transport of carbon dioxide within soils is also an important process potentially responsible for hysteresis between $R_s$ and $T_s$. High soil water content tends to block $CO_2$ gas transport, resulting in large diel hysteresis between $R_s$ and $T_s$ (Riveros-Iregui et al., 2007). However, from our results, the diel hysteresis between $R_s$ and $T_s$ decreased with increasing *SWC*. Thus, carbon dioxide transport within soils cannot fully explain the variation that we see in our measurements. Therefore, our results preclude placing too much importance on gas transport processes on seasonal variation in hysteresis at our site.

Transportation of carbon dioxide within the phloem likely had some influence on both the magnitude and time trending of $R_s$. It should be taken seriously in the study of diel hysteresis. However, we do not currently possess the instrumentation that would allow us to make continuous measurements of carbon dioxide within the phloem. Also, it is rather difficult to make such measurements on small plants, such as *Artemisia ordosica*. Transportation of carbon dioxide within the phloem is still a new research area with many issues remaining unclear. It will be taken into account in our future research.

Currently, no consensus has been reached on the causes of diel hysteresis. We are not clear if hysteresis stems as a result of gas or photosynthate transport processes, or from both. In this context, we selected a study site of sandy soil to weaken the influence of gas transport processes and a small *Artemisia ordosica* plant to minimize the influence of photosynthate transport. Interestingly, our results still show large diel hysteresis between $R_s$ and $T_s$. Based on our analysis and earlier studies by other researchers, we hypothesize that *SWC* has a role in regulating the relative control between photosynthesis and temperature on diel $R_s$ by changing the relative contribution of autotrophic and heterotrophic respiration to total $R_s$. Our arguments may not be generalizable to the level that the editor may be searching for, but they provide impetus for future studies along this line of inquiry.

**Specific comments:**
The text throughout needs to be more efficient. One example is "biologically photosynthesis-related processes". This could also be written "photosynthesis". Every sentence in the manuscript could be made simpler and more effective.
**Response:** We accept the criticism and tried to do this as much as we could; we have changed "biologically photosynthesis-related processes" to "photosynthesis", and

"biological-based" to "biological" throughout the manuscript, see page 3, line 7, 13-15, and Page 1, Line 23.

On page 1 line 10, I do not feel that diel hysteresis between respiration and temperature is controversial, rather that both biological and physical mechanisms can explain it (see for example Stoy et al. 2007, http://onlinelibrary.wiley.com/doi/10.1111/j.1365-3040.2007. 01655.x/full, and note also Detto et al. for a very nice time series treatment of this challenge http://www.journals.uchicago.edu/doi/abs/10.1086/664628).
**Response:** We accept and changed this sentence to "No consensus has been reached on the causes for diel hysteresis between soil respiration ($R_s$) and temperature". See page 1, line 10.

The abstract describes the findings succinctly, but the notion that autotrophic respiration is responsible for the changes is merely a hypothesis. Note that this is an important area of future research rather than stating 'it is indicated', which lends false confidence to the findings.
**Response:** We accept. We deleted the sentence "by changing the relative contribution of autotrophic and heterotrophic respiration to total $R_s$". We added "We recommend further studies to explore the actual mechanisms involved in explaining changes in the relative contribution of autotrophic and heterotrophic respiration to total $R_s$. These studies may help elucidate the role of $SWC$ in affecting seasonal variation in diel hysteresis." at the end of abstract. See page 1, line 20-22.

All of the statements that follow "e.g" in the second paragraph of the introduction can easily be argued against and should be clarified or quantified. Note Reichstein et al. 2003 (doi: 10.1029/2003GB002035) regarding statement on line 9 of page 2.
**Response:** We accept. We changed the sentence to "Generally, soil $CO_2$ fluxes are measured at the soil surface and related to temperatures in the soil. Transport of $CO_2$ gas to the soil surface takes time to occur, which may cause delays to appear in observed respiration rates, causing hysteretic loops to form between $R_s$ and $T_s$ (Zhang et al., 2015).''. See Page 2, line 9-11.

The statement "Some studies have reported that high SWC tends to block $CO_2$" should be strengthened by noting the basic equations for transport in porous media (e.g. Millington and Quirk) demonstrate very convincingly that it does on theoretical grounds, not merely via empirical findings.
**Response:** We accept and change the sentence to "Based on the Millington-Quirk model, high $SWC$ block $CO_2$ gas and thermal diffusion (Millington and Quirk, 1961)". See page 2, line 24-25.

The statement "few previous studies address soil respiration dynamics in drylands" is not correct. Many have.

**Response:** We accept and deleted the text "However, compared to forest ecosystems, few previous studies address soil respiration dynamics in drylands". See page 2, line 31.

The statements on page 3 lines 7-9 likely would not hold in a system with biological crusts that may hinder gas transport.

**Response:** We do believe that biological crusts have effects on gas transport. As reported by Feng et al. (2014), biological soil crusts increase the magnitude of the lag. However, biological soil crusts likely have little effects on the variation of the lag between $R_s$ and $T_s$. For example, the lags changed day to day, whereas the influences of biological crusts on gas transport did not. Therefore, we change this sentence as "Therefore photosynthesis may govern the level of variation in asynchronicity between $R_s$ and $T_s$ in drylands". See page 3, line 7-8.

"The soil at the research station was of a sandy type" is an insufficient description of the soil in this case in which soil properties are critical for understanding results.

**Response:** We added the information of total porosity as "The total soil porosity within 0-2 cm and 5-25 cm depth was 50% and 38%, respectively." See page 3, line 29.

How was the stabilization period determined?

**Response:** Generally, according to previous studies, the stabilization period was range from several days to several weeks. In our case, the period of March-May (inclusive) was considered as the stabilization period. We start to record the flux data in June after three months since measurement collar installation. To avoid such confusion, we deleted the text "after an initial period of site disturbance stabilization from March, 2012 to June, 2012". See page 5, line 15-16.

The text surrounding equations 1-3 is confusing; please rewrite for clarity. The description for equation 3 is inaccurate as it refers to the instantaneous importance of different variables for respiration rather than their actual effect, which may be lagged.

**Response:** The text has been rewritten. See Page 4, line 29 – Page 6, line 8.

The following statement wasn't substantiated is qualitative, it needs to be rewritten: "However, sandy soils with high soil porosity, as on our site, have a minor influence on within-soil gas transport processes." Using an actual physical model of these processes and differences among soil types and moisture contents would improve the manuscript dramatically; at the moment the arguments are empirical and quite frankly often times rather weak.

**Response:** We accept and changed the text to "However, at our site, sandy soils with high porosity, $SWC$ was lower ($< 0.15$ $m^3$ $m^{-3}$, Fig 1). According to the Millington-Quirk model (Millington and Quirk, 1961), changes in $SWC$ may have minor influences on within-soil gas transport processes.". See page 7, line 10-11.

There are multiple alternative hypotheses to the statement on page 7 line 14 that the dominant respiratory source changes with depth and/or that respiration is suppressed at high temperatures in desert ecosystems (see for example http://link.springer.com/article/10.1007/s10533-010-9448-z).

**Response:** Although multiple alternative hypotheses were used to explain the suppressed $R_s$ under high temperatures, according to Figure 2, our results directly suggest that the suppressed photosynthesis under high temperature and low soil water content was likely the reason for suppressed $R_s$, as the diel trend of $R_s$ followed the diel variation in photosynthesis.

Very surprised that section 4.2 excludes a discussion of the Birch effect (e.g. Jarvis et al. 2001, http://www.geosciences.ed.ac.uk/homes/lwingate/publications/Jarvis_Birch.pdf).

**Response:** We have cited the paper and added the Birch effect in our discussion (see Section 4.2).

The statement "Most organic matter and microbes tend to concentrate in the upper part of the soil, whereas plant roots are found much deeper" isn't necessarily true.

**Response:** According to our field measurements, which we do not show in the study, over 80% of total soil organic matter and soil nitrogen is found in the top 0-10 cm of the soil . Beyond our measurement, Ciais et al. (2011), Thomas (2012) and Gao et al. (2014) also reported such results. Thus, we expected it to be true for our study; the surface of the soil is a source of these elements. We added the text "At the study site, over 80% of soil organic matter and soil nitrogen was concentrated in the first 10 cm of the soil." in site description part. See Page 3, line 30.

As noted also by the referees, the statements regarding heterotrophic versus autotrophic respiration are merely speculative.

**Response:** Without heterotrophic and autotrophic respiration data, we cannot clearly show the relative changes in the contribution of heterotrophic and autotrophic respiration to total $R_s$. The birth effect (Jarvis et al., 2001) and the study by Casals et al. (2011) show that the relative changes in autotrophic respiration and heterotrophic respiration respond to changes in *SWC*. Such changes are likely a result of changes in relative control between photosynthesis and temperature on $R_s$, which is consistent with our own results, Figure 4. We think our speculation is reasonable.

The statement "Our work provides urgently needed new knowledge concerning causes/mechanisms" isn't accurate. A mechanistic study of the processes controlling soil respiration were not undertaken; the analysis is entirely empirical. The following statement that a soil moisture threshold of 0.08 is of particular importance is not likely to hold in different desert systems with different soil properties and vegetation.

**Response:** We accepted and delete "concerning causes/mechanisms". We agree that soil moisture threshold of 0.08 is of particular importance for our study area. To make

a general description, according to Referee 2, we added the text '(ratio of *SWC* to soil porosity = 0.26)' behind the threshold (0.08 $m^3$ $m^{-3}$) in Introduction (Page 1, line 17), Results (Page 6, line 19), Discussion (Page 7, line 19) and Conclusions (Page 8, line 18) paragraphs in the manuscript.

[revised manuscript text omitted]

---

## Author Response (AR2)

**Responses to editor's comments**

We appreciate the editor's thoughtful and constructive comments. We have fully considered the editor's comments in the revision of the manuscript. Response to the editor's questions appear in blue.

Comments to the Author:

I thank the authors for their careful responses. The edited manuscript was difficult to read in places due to the pdf markup used (I couldn't figure out how some changes were being resolved nor read added text). A draft in which additions to the text can be identified would be appreciated.

**Response:** Sorry for that error, a draft with clear track changes has been submitted with corresponding response. Hopefully, this will make the reading easier.

My concern with the manuscript is that I still don't agree with equation 2 given the necessary lag between photosynthesis and respiration, which is likely to increase under drought stress depending on the response of phloem velocity to changes in xylem velocity. These terms will change as a function of drought stress. There is also of course the argument root-to-soil carbon flux occurs shortly after photosynthetic inputs to the plant due to osmoregulatory constraints, but it is unclear how large this effect is. Work by Mencuccini and others may add insight into time lags of transport between photosynthesis and respiration; the response in the letter regarding these dynamics was too terse. In brief, I feel that this manuscript is confirmatory and makes some interesting points regarding seasonal changes in photosynthesis/respiration coupling. At the same time, it is far less rigorous than many manuscripts that address similar issues, hence my concern that it is not sufficiently advancing the science. In summary, I would appreciate it if the authors could send a draft in which changes to the manuscript were more clear, and cite more relevant literature on photosynthesis/respiration coupling, which they might have but again these additions from the pdf that was uploaded is unclear.

**Response:** we agree with you that Eq. 2 may incorporate some bias in the description of the relationship between photosynthesis and soil respiration due to lag. However, we still found Eq. 2 to fit 24% of the days with diel $R_s$ with no observable lag over the growing season. Thus, in order to improve Eq. 2, we needed to understand the lag that existed between photosynthesis and soil respiration. This is one of the aims of the study. Further analysis shows that the $R_s$-function with photosynthesis and temperature as explanatory variables (Eq. 3 in the text) was able to fit 84% of the days with no observable hysteresis. In addition, the calculated hysteresis between photosynthesis and soil respiration using Eq. 2 is the same as that based on the observed data. Therefore, we think Eq. 2 can be used.

More details about the influence of *SWC* on photosynthate transport in phloem has been added to the second part of the discussion of this manuscript.

In order to provide greater support for our conclusions, new results and discussion related to the influence of photosynthesis on diel $R_s$ and seasonal variation in diel hysteresis have been added to the manuscript.

Relevant literature on photosynthesis-soil respiration coupling has been added to the manuscript and reference list.

A couple of minor notes:
Regarding the statement 'rather than rely on relating daytime simulations of Rs to night-time respiration-temperature relationship', note that many partitioning approaches don't, but I agree that it is the normal course of affairs.

**Response:** we deleted the statement "This information would be particularly useful when processing ecosystem fluxes obtained with eddy covariance measurements, rather than rely on relating daytime simulations of $R_s$ to night-time respiration-temperature relationships. This would involve more complex, iterative methods than are currently used because of the implied feedback."

'No consensus has been reached' implies that there is a single cause of hysteresis. But there are many, and different ecosystems should behave somewhat differently with respect to this hysteresis.

[revised manuscript text omitted]

---

## Author Response (AR3)

**Response to editor's comments**

We appreciate the editor's thoughtful and constructive comments. We have fully considered the editor's comments in the revised manuscript. Responses to the editor's comments appear below in blue.

Comments to the Author:
The manuscript represents an improvement, but the authors still seem to be affixed to the idea that there is a discrepancy in the literature regarding physical vs. biological controls over soil respiration and erroneously describe manuscripts in the literature as 'contradictory' when no contradiction exists. Physical and biological controls are both important, and their degree of control will differ in different ecosystems. By reframing the manuscript to actually describe the findings of the manuscripts that the authors incorrectly identify as 'contradictory' (see for example page 9 line 5. Eight citations without describing a single one!!!), the authors will be able to contribute to the literature rather than create an artificial debate where little debate exists. There is still an alarming lack of mechanistic reasoning for key passages in the manuscript as described in more detail below. Please note the approach and conclusions of Detto et al. (doi: 10.1086/664628) as it applies to photosynthesis/respiration coupling as you see fit.

**Response:** We agree that both biological and physical processes are important in the control of hysteresis between $R_s$ and $T_s$. The manuscript has been reframed to reflect this point.

Our manuscript mainly focuses on the relative degree of control between physical and biological processes on hysteresis. After this study, our next work is to study the coupling of photosynthesis and respiration in desert shrubland. Thanks for the suggestion of Nonparametric Spectral Granger Causality Approach (Detto et al., 2012), which is better than classical correlation analysis with detecting causality in time series data. A second approach, Convergent Cross Mapping (CCM, Sugihara et al., 2012, DOI: 10.1126/science.1227079), may also be useful with detecting causality in complex dynamic systems. Both approaches will be considered in a new paper.

In the title write 'desert shrubland'.
**Response:** We agree. We have changed 'desert-shrub land' to 'desert shrubland' in the revised manuscript (line 2-3).

I basically disagree with line 12. The implications of Stoy et al. (2007) and other manuscripts is that physical and biological mechanisms cannot be fully separated from photosynthesis / soil respiration measurements because the time scales of biological and physical transport processes are concomitant (see also page 3 line 12). That being said, I agree that there is a substantial role for biology in determining hysteresis. In other words, when you write "Currently, it is not clear whether physical or biological processes (or their combination) dominate the control of diel hysteresis in drylands" it is in fact clear. Both do. The relative degree of control of each is what is of more interest.

By framing the manuscript with respect to the degree of control of physical and biological processes, rather than trying to find out which is the control when both are important, would lead to a much less questionable argument to motivate the analysis.

**Response**: We agree. The second paragraph in the introduction was reframed (lines 37-53). The sentence was changed to 'it is not clear to what degree physical and biological processes control hysteresis in drylands.' (line 67-68)

On page 2 line 16 (from here on out please always only use continuous line numbering when preparing manuscripts) one might argue that substrate in addition to temperature are equally important for determining soil respiration.

**Response:** Continuous line numbering has been used in the revised manuscript. We agree that substrate in addition to temperature are important for determining soil respiration. We changed 'but are for the most part' to 'and are' to reduce the argument (line 40). The influences of substrates to $R_s$ has been described as photosynthate supply in the text (line 42-50).

There is no information to support the following statement: "At our study sites, it is likely that $R_s$-effluxes at the surface originated from biogeochemical processes in the deep soil." There is also no reason to frame the references in the previous sentence (p9 L5) to be 'contradictory'. The authors still seem like they are trying to solve some challenge in science with respect to physical vs. biological controls over soil respiration. Because both are important, this isn't a particularly useful way of thinking about the processes at hand.

**Response:** We agree, both sentences have been deleted from the text (line 203-206).

On page 9 line 20, please describe how these processes could occur on the order of minutes in sandy soils.

**Response:** We agree. Based on our latest revision, the manuscript focuses on the degree of control of photosynthesis on hysteresis. Thus, we have deleted discussion on the control of physical processes, due to lack of data and analysis (line 214-219).

Please cite Jorge's name as "Curiel Yuste" correctly on page 11 line 20.

**Response:** We agree. Jorge's name has been correctly revised in the manuscript (line 255 and 302).

[revised manuscript text omitted]